# Hereditary Colorectal Cancer: State of the Art in Lynch Syndrome

**DOI:** 10.3390/cancers15010075

**Published:** 2022-12-23

**Authors:** Antonio Nolano, Alessia Medugno, Silvia Trombetti, Raffaella Liccardo, Marina De Rosa, Paola Izzo, Francesca Duraturo

**Affiliations:** 1CEINGE Advanced Biotechnologies Scarl, “Francesco Salvatore” Napoli, Department of Molecular Medicine and Medical Biotechnologies, University of Naples Federico II, 80131 Naples, Italy; 2Department of Molecular Medicine and Medical Biotechnologies, University of Naples Federico II, 80131 Naples, Italy; 3Department of Veterinary Medicine and Animal Productions, University of Naples Federico II, 80137 Naples, Italy

**Keywords:** Lynch syndrome, MMR genes, VUS MMR genes, MSI-status, molecular diagnosis, Lynch-like syndrome

## Abstract

**Simple Summary:**

Lynch syndrome is the most common form of hereditary colorectal cancer associate to variants in *Mismatch Repair* (*MMR*) genes. Unfortunately, a large amount of variants identified in these genes remain of uncertain significance. Therefore, many individuals with a clinical suspicion of LS receive a diagnosis of Lynch-like syndrome. This review summarizes the main aspects of Lynch syndrome and recent advances in the molecular diagnosis and, in particular the main factors that determine the loss of expression of *MMR* genes.

**Abstract:**

Hereditary non-polyposis colorectal cancer is also known as Lynch syndrome. Lynch syndrome is associated with pathogenetic variants in one of the mismatch repair (MMR) genes. In addition to colorectal cancer, the inefficiency of the MMR system leads to a greater predisposition to cancer of the endometrium and other cancers of the abdominal sphere. Molecular diagnosis is performed to identify pathogenetic variants in MMR genes. However, for many patients with clinically suspected Lynch syndrome, it is not possible to identify a pathogenic variant in MMR genes. Molecular diagnosis is essential for referring patients to specific surveillance to prevent the development of tumors related to Lynch syndrome. This review summarizes the main aspects of Lynch syndrome and recent advances in the field and, in particular, emphasizes the factors that can lead to the loss of expression of *MMR* genes.

## 1. Introduction

Lynch syndrome (LS) is the most common form of hereditary colorectal cancer, with an incidence of between 2% and 3% of all colorectal cancers (CRCs) [1], followed by familial adenomatous polyposis (FAP), which accounts for less than 1% of total CRCs [2] and other inherited syndromes, such as hamartomatous polyposis [3], Table 1. LS is also known as hereditary non-polyposis colorectal cancer (HNPCC); however, colorectal cancer develops due to a malignant transformation of adenomatous polyps, but they are not numerous and widespread as instead observed in FAP, which is characterized by 100–1000 polyps [4].

Although the incidence of early-onset colorectal cancer, which occurs in individuals <50 years of age, has been increasing worldwide and particularly in high-income countries [5], LS patients generally develop colorectal cancer at an early age (on average about 45 years), with a predominance of 70% in the proximal/right colon [6].

Affected patients also present with synchronous tumors (multiple malignant tumors) and metachronous tumors (the appearance of a second tumor in one or more colorectal segments in patients who have already undergone resection surgery for cancer).

The precursor lesion of CRC in individuals with LS is an adenoma, which occasionally may be flat rather than raised or polypoid. Compared to patients with attenuated polyposis syndromes, patients with LS develop fewer colorectal adenomas by 50 years of age (usually less than three neoplasms) [6]. Colorectal adenomas in patients with LS exhibit accelerated carcinogenesis, leading to transition to carcinoma within 2 to 3 years, in contrast to the 8 to 10 years this process may take in the general population [7].

In addition to CRC, patients with LS have a significantly increased risk for a wide variety of cancers in other body sites, such as the endometrium, ovary, stomach, small intestine, hepatobiliary tract, pancreas, urinary tract, prostate, brain, and skin [7,8].

CRC associated with LS has clinical features distinct from those of sporadic CRC, often showing a combination of the presence of prominent tumor-infiltrating lymphocytes with marked lymphocytic inflammation that resembles the “Crohn’s-like reaction,” poor differentiation, and presence of mucinous and/or ring-like cells [9,10].

Although fewer studies have been published on non-colorectal LS-associated cancers, LS-associated endometrial cancers may be seen more frequently than their sporadic counterparts in the lower uterine segment; the majority are of the endometrioid type and often show poor differentiation, with tumor-infiltrating lymphocytes [11].

A systematic literature search was conducted in the literature to identify the studies describing clinicopathologic characteristics, functions of MMR system, variants in MMR genes and genotype–phenotype correlations, MMR protein immunohistochemistry and/or MSI, genetic testing in Lynch syndrome cancer patients, and the studies on Lynch-like syndromes. In this manuscript, the information from both original articles and reviews were reported.

## 2. MMR Genes

LS is inherited in an autosomal dominant fashion and develops due to a germline mutation in one allele of one of the DNA MMR genes.

In the human mismatch repair (MMR) system, MSH2, MSH3, and MSH6 proteins associate in two heterodimeric complexes as MSH2-MSH6 (MutSα) and MSH2-MSH3 (MutSβ), which is homologous to the bacterial MutS protein [12], Table 2.

On the one hand, the first complex is able to recognize and bind DNA at the site of a mismatch due to substitution, insertion, or deletion of a single base. On the other hand, the second complex is responsible for the identification of insertions or deletions of a few nucleotides (2–4 bases). MSH2 protein is essential for the functional constitution of both complexes.

The heteroduplex formed by MLH1 and PMS2 (MutLα) or by MLH1 and MLH3 (MutLγ) interacts with the MutSα or MutSβ complex and stimulates the excision and resynthesis of DNA [13]. As already pointed out for the role of MSH2 protein within the MutSα–MutSβ complex, MLH1 protein is essential for the functional constitution of the MutLα and MutLγ complexes.

As a result, the MutLα–MutLγ complex coordinates the reciprocal action between the “mismatch” recognition complex and the other proteins necessary for the excision and resynthesis of the wrong strand. These additional proteins include DNA polymerases δ and ε (Polδ and Polε), the proliferating cell nuclear antigen factor (PCNA), an exonuclease (EXO1), and a replication factor C (RFC), Table 2.

The ATPase activity of the MutSα complex is important for the interaction with the unpaired DNA and the initiation of repair activity. MutSα binding stimulates the hydrolysis of ATP, leading to a conformational change that consequently triggers the recruitment of the MutLα complex. The tetrameric complex moving along the DNA looks for the mismatch present on the newly synthesized strand, which in turn activates the PCNA factor and RFC. MutLα possesses intrinsic ATP-mediated endonuclease activity, which is activated by PCNA. This activation causes an incision in the newly synthesized strand containing the error. This is followed by the recruitment of EXO1, which removes the newly synthesized strand containing the pairing error, in order that the strand can be synthesized again by DNA polymerase δ, while ligase 1 joins the previously created ends [14,15], Figure 1.

## 3. Other Functions of MMR Genes

In addition to fulfilling their role in repairing DNA damage, MMR proteins perform other highly relevant functions in carcinogenesis [13]. As shown in Figure 2, these roles include:prevention of reparative recombination (gene conversion) between non-identical sequences [16];promotion of meiotic cross-over, which involves the MLH1, PMS2, and MLH3 proteins in particular [16,17];protection against intergenerational instability resulting from the phenomenon of trinucleotide repeat expansion, which is the basis of the pathogenesis of various neurodegenerative diseases [18];the immunoglobulin (Ig) differentiation process based on “somatic hypermutation”, regulated by the MutSα–MutLα complex in combination with two other proteins, AID (activation-induced cytidine deaminase) and Polμ (error-prone DNA polymerase) [19];modulation of microRNA (miRNA) biogenesis through the interaction of MMR proteins with the microprocessor complex; in particular, MutLα specifically binds to pri-miRNAs and the Drosha–DGCR8 complex to stimulate the processing of pri-miRNAs into pre-miRNAs in a manner dependent on the ATPase activity of MutLα [20];reporting of DNA damage caused by exogenous carcinogens (heterocyclic amines, oxidizing agents, and UV radiation) obtained through a synergistic action between the homologous proteins of p53 (p53, p63, and p73) and the MutSα–MutLα complex. Moreover, in response to exogenous damage, MLH1 interacts with the MRE11 protein, a component of the BRCA1-associated surveillance complex (BASC), and regulates the cell cycle and the apoptotic pathway; indeed, there is a correlation between the MMR system and the G2/M phase of the cell cycle [21].

In particular, during carcinogenesis, the apoptotic mechanism is deregulated and thus the cells tend to escape the programmed death process and to bypass any cellular damage. Therefore, cells cannot fulfill their normal growth function if a mutation is present in some genes related to tumor development, and the MMR system may be representative of only this scenario. Indeed, several studies show how the MMR system plays an important role in the apoptotic machinery and in the activation of cell cycle check points [21,22,23]. Among the MMR genes, MLH1 and MSH2 are, above all, the most studied in relation to the anomalies found in apoptotic processes. In particular, the MSH2 gene plays a key role in genomic stability. In addition to its DNA damage repair function, it acts as a “sensor” for DNA replication errors caused by DNA base analogs and binds to various damage-induced DNA adducts to trigger cell cycle arrest or apoptosis [13,21,22,23].

The combination of all of these functions makes MMR proteins extremely important in maintaining the integrity of the genetic material, in the regulation of the cell cycle, and in the development of an effective immune system. Consequently, when the functionality of the MMR is lost or defective, there is a decrease in apoptosis and an increase in cell survival and mutagenesis induced by the damage, which leads to a selective growth of the defective cells with a parallel increase in tumor susceptibility.

## 4. The Molecular Alterations of Lynch Syndrome 

Patients with LS have a germline mutation in one of the MMR genes. Each MMR protein encoded by the corresponding gene has a unique function in repairing replication errors. Therefore, MMR proteins possess unique functional domains. When mutations occur in the site corresponding to the functional domain, the DNA repair function can be impaired [24,25].

The International Society for Gastrointestinal Hereditary Cancer (InSiGHT) was the first group of experts to define the pathogenicity of gene variants according to an agreed set of criteria based on Bayesian probability, using the five-level classification system. In this system, the variants of Class 5 are pathogenic and Class 4 are possibly pathogenic; in Class 3 the variants are of uncertain significance (VUS), and Classes 2 and 1 variants are identified as probably benign and benign, respectively [26,27]. In the InSiGHT database, (www.insightdatabases.org, 01 April 2022) which contains more than 3000 variants associated with MMR genes, 82% of the listed variants of Classes 4 and 5 involve MLH1 and MSH2; 13% involve MSH6 and 5% PMS2 [28,29]. In contrast, with regard to other genes, few pathogenic mutations have been found in MLH3, while only a heterozygous variant in the MSH3 gene has been associated with LS [30,31].

The percentage distribution above can be explained by the fact that MLH1 and MSH2 are the most important predisposing genes for LS. Their protein products are mandatory components in all types of heterodimers of the repair system [12]. Most mutations that fall into MLH1, MSH2, PMS2, and MSH6 are truncating, predominantly nonsense, or frameshift mutations [32]. The proportion of missense mutations leading to single amino acid substitutions is between 30% and 60% for all four genes [33]. About 60% are variants that fall into the regulatory regions of splicing, which is potentially altered and causes a truncating effect in the protein [34]. Among the variants cataloged as pathogenic, there is every type of alteration, including nonsense, missense, or frameshift mutations, mutations that alter the splicing sites, insertions, or deletions of one or a few exons, and large rearrangements, of which the largest found to date has been a cytogenetically visible 10 Mb inversion affecting MSH2 [35,36]. Non-synonymous missense changes make up about 68% of Class 3 variants, i.e., VUS [23]. The share of VUS is 31% for all variant showed for MLH1, 28% for MSH2, 47% for MSH6, and 26% for PMS2 [37], Figure 3.

## 5. Variants of Uncertain Significance

Although the variants that determine the production of a truncated protein predominate among the alterations that cause a hereditary pathology at the germinal level, sequencing analysis has revealed that about 40% of patients who are probable carriers of an inherited tumor syndrome have a variant of uncertain significance (VUS) [38]. These variants typically comprise a single amino acid substitution that a priori cannot be classified as pathogenetic or benign [39].

Therefore, the detection of a VUS suggests a critical scenario since, in the pathogenicity–benignity spectrum, it is not known exactly where this type of variant can be placed. In addition, the status of a VUS carrier does not stratify the members of a family into those at higher or lower risk [35].

Therefore, the influence of these variants on cancer development is a controversial topic [40,41]. The key analytical problem is that the effects of a VUS must not be categorically “all or nothing”. Regardless of the type of method used for their detection, it is difficult to determine what proportional reduction in normal function, due to an abnormal protein, is necessary to confer an increased and clinically relevant tumor risk [38], Table 3.

However, even a synonymous nucleotide substitution, which normally does not cause an alteration in the production of the protein, can be pathogenetic if, for example, it impairs splicing [41]. Furthermore, two or more variants (of the same gene or in different genes) may coexist and even co-segregate with the disease phenotype in a single family, and this could be compatible with different interpretations, including a real correlated predisposition. 

In some cases, a VUS can make a more concrete contribution to cancer risk than classic pathogenetic Mendelian variants. In addition, the simultaneous presence of a VUS and some polymorphisms in different genes that are implicated in tumor predisposition, while behaving as low-risk alleles, could cooperatively contribute to an increased risk of cancer development [31].

Therefore, recent studies have shown that a significant percentage of hereditary susceptibility related to the most common diseases may be the result of an additive effect of a series of low frequency variants of different genes. These low risk alleles probably act in a dominant and independent way, giving each of these a moderate, but detectable, increase in the relative tumor risk [30,31,42,43].

## 6. Microsatellite Instability (MSI)

Deficiency of the MMR complex results in a high rate of mutations in repetitive DNA sequences known as microsatellites. Microsatellites are short 1-6 base DNA motifs that are repeated and distributed throughout the genome in both coding and non-coding regions. Due to their repeating structure, microsatellites are particularly prone to replication errors which are normally fixed by the MMR system. The loss of function of one of the MMR proteins causes an accumulation of errors in the microsatellites, with insertions or deletions of bases, resulting in genetic instability [44].

Microsatellite instability (MSI) can have an oncogenic potential when it occurs in the coding regions of genes involved in various crucial cellular functions and pathways [45]. More than 30 genes have mutations that occur in microsatellite repeats in MMR-deficient tumors and these genes are implicated in several cellular functions and pathways. Examples are the DNA repair proteins MRE11A and RAD50, the growth factors TGFBRII and IGFRII, the pro-apoptotic factor BAX, the same MMR genes *MSH3* and *MSH6*, and the histone modifier *HD2* [46]. Currently, for the detection of microsatellite instability (MSI), a panel composed of five mononucleotide microsatellites is used. Tumors showing instability for at least two of these repeats are classified with high instability (MSI-H); those with instability in a repeat are classified with low instability (MSI-L), while tumors without alteration are classified as stable (MSS). In addition to the evaluation of the state of microsatellite instability, the analysis of the expression of MMR proteins (MLH1, MSH2, MSH6, PMS2) by immunohistochemistry (IHC) permits the diagnosis of their possible lack of expression and enables the identification of patients with suspected Lynch syndrome (LS).

Analysis using IHC represents a specific (100%) and sensitive (92.3%) screening tool for identifying MSI-H tumors [47,48]. Therefore, through IHC and/or MSI testing, it is possible to direct people with suspected LS toward the molecular test used to search for mutations in the *MMR* genes and to confirm the clinical suspicion of LS in patients, as well as to conduct a pre-symptomatic molecular diagnosis in the context of at-risk family members.

The MSI phenotype is found in approximately 90% of Lynch tumors and only in 15% of sporadic colorectal cancers (CRCs) [48]. In the latter case, the unstable condition is mainly caused by somatic hypermethylation of the promoter of the *MLH1* gene [45].

## 7. K-Endometrium in LS and Loss of MMR Proteins

Endometrial cancer associated with LS is the most common extracolic tumor form in the context of tumors that fall within the Lynch spectrum. The likelihood of endometrial cancer development as the first neoplastic manifestation is about 40–60% in women with LS [49]. It is defined as a “sentinel cancer” associated with LS [50]. Therefore, this definition identifies the risk of the subsequent development of other cancers associated with LS and thus the need for early screening and preventive strategies to decrease cancer-related morbidity and mortality. Recent studies that have identified and deepened the knowledge regarding the loss of expression of MMR proteins have focused on non-neoplastic colonic crypts [51,52]. This type of evaluation, linked to the expression of the proteins of the MMR system, was recently conducted in a cohort of patients with endometrial cancer and provided a further application useful for the screening of LS, as well as being able to guarantee a better understanding of the pathogenesis of LS-associated endometrial cancer [53].

However, literature data on the prevalence and potential significance of MMR deficiency in pre-cancerous endometrial lesions are rather limited. In a small cohort of patients with LS, loss of expression of MMR proteins, as defined by IHC and/or MSI testing, was found in areas of endometrial hyperplasia adjacent to endometrial cancer [54,55,56]. In one of these studies, a higher frequency of MSI and an earlier mean age of onset of cancer were found in carriers of the MSH2 gene mutation than in carriers of mutations in other MMR genes, suggesting that an MSH2 mutation may indicate a faster rate of tumor progression [54].

Details concerning the molecular pathogenesis of endometrial carcinoma in LS are not yet fully known: It is conceivable that non-neoplastic endometrial glands with MMR deficiency may represent the initial step in endometrial carcinogenesis in patients with LS, leading to the development of atypical hyperplasia or a direct evolution to carcinoma.

## 8. Loss of Expression of MMR Genes

Many hypotheses have been proposed concerning additional factors that may result in a loss of function in the MMR system, alongside better known and more characteristic alterations of the gene repair pathway.

The mutator phenotype is the result of incorrect functioning of the MMR system. In addition to the presence of germline mutations, various pathogenetic events, including methylation of the *MLH1* promoter [57] and reduced histone acetylation [58], lead to reduced or absent expression of MMR proteins, as well as factors related to the microenvironment, such as inflammation and hypoxia [59,60].

The maintenance or non-integrity of MMR proteins, including the performance of their functional role, can also be controlled by miRNAs; namely, non-coding RNAs that canonically play a role in post-transcriptional regulation of the control of critical biological processes, including development, differentiation, cell proliferation, and apoptosis [61,62].

For example, it was observed that miR-155 overexpression was present in CRC [63,64,65] and appears to be more frequent in MSI than in CRC MSS [66]. These data are consistent with the hypothesis that miR-155 can regulate the components of the MMR machinery and, consequently, the mutation rates and MSI. The miR-155 leads to downregulation of the main heterodimeric proteins of the MMR; namely, MSH2-MSH6 and MLH1-PMS2, generating a mutator phenotype [67]. It alters both the expression and the stability of the MMR pathway at somatic level, with a consequent significant increase in mutation rates. In support of these observations, an inverse correlation has also been found between miR-155 and the expression of MMR proteins in CRC cells.

In general, alterations in the miRNAs that regulate MMR can be caused by mutations in the target sequences, as demonstrated by the presence of acquired mutations in the 3’UTR of the *MLH1* gene, for miR-422a [20], or mutations in the 3’UTR of the *MSH2* gene, for miR-137 [68].

Therefore, it should always be considered that variants in some gene regions, such as in the 3’UTR, could compromise the binding of putative transcription factors or miRNAs involved in the regulation of gene expression, Figure 4.

## 9. Genotype–Phenotype Correlations

The risk of cancer development during life is significantly higher in carriers of *MSH2* and *MLH1* mutations than in carriers of *MSH6* or *PMS2* mutations. The lifetime risk of any LS-associated cancer up to age 70 varies between 57% and nearly 80% for carriers of *MSH2* mutations and between 59% and 65% for carriers of *MLH1* mutations [69,70]. For *MSH6*, the risk is 24% for men and 40% for women [71] while the risk of cancer development throughout life varies from 25–32% for carriers of a *PMS2* mutation [72].

Among the various cancers that arise in carriers of mutations in *MSH2* and *MLH1*, the highest risk is clearly related to the onset of CRC over the course of life, followed by endometrial cancer and other extracolic cancers. Moreover, individuals with alterations in the *MSH2* gene show a higher incidence of other extracolic manifestations (gastric and renal cancer) than those with mutations in the *MLH1* gene [73].

Mutations in the *MSH6* gene, for example, appear to cause an “attenuated” form of hereditary non-polyposis CRC, characterized by a lower penetrance, a higher age of onset of the disease, and a low degree of microsatellite instability (MSI-L) [29]. Women carrying *MSH6* mutations have a higher risk of endometrial cancer than CRC [74].

Defects in the *PMS2* gene also cause a milder phenotype, with distinct characteristics of tumors caused by mutations in *MLH1* and *MSH2*; compared with *MSH6*, they demonstrate early development of the tumor, which shows MSI. In addition, the *MSH6* and *PMS2* mutations exhibit reduced penetrance, resulting in a higher mean age at onset of various tumors in their carriers than in those with the *MSH2* or *MLH1* mutation [75]. Regarding minor genes, mutations in the *MLH3* gene have been associated with brain tumor development [30]. Variants of the *MSH3* gene have been associated with a classic phenotype only if they are inherited together with variants of the *MSH2* gene [31]. Furthermore, biallelic mutations in the *MSH3* gene have been shown to cause a polyposis form similar to the FAP phenotype [76].

This heterogeneity also occurs among family members who share the same mutation. Indeed, other factors, such as environmental or polygenic factors, can influence the phenotypic expression. In this context, counseling and the planning of surveillance strategies should be tailored to each patient from the perspective of age, sex, and the pathological variant identified.

## 10. Lynch-like Syndrome

In approximately 60–70% of cases where LS is clinically suspected but genetic testing has failed to identify a germline mutation in the *MMR* genes, patients are defined as having Lynch-like syndrome (LLS) [77,78]. Similar to those with LS, patients with suspected LLS similarly present with cancer at a young age (53.7 vs. 45 years) and manifest microsatellite instability (MSI), as well as immunohistochemical absence of an MMR protein [79,80]. The analysis of tumors between probands and families demonstrates heterogeneity for the risk of tumor onset, with a lower incidence for CRC in LLS syndrome than in LS, as well as for tumors in other locations [81].

There are several potential explanations for LLS. First, it is possible that some LLS patients may actually have Lynch syndrome, as there may be some germline mutations in the *MMR* genes that are not detectable by current genetic testing, such as those that fall into intronic and promoter sequences [82]. An alternative explanation is that there are other molecular mechanisms that inactivate MMR in addition to canonical inactivation via a gene mutation; indeed, MMR activity may also be modulated by changes in MMR gene expression, causing the same tumor phenotype that closely resembles Lynch syndrome [81]. Finally, recent findings suggest other molecular mechanisms, such as miRNAs, that could control *MMR* genes expression and determine its dysregulation in promoting tumorigenesis in colon cells [82]. Moreover, it should be noted that in contrast to sporadic microsatellite instability-high (MSI-H) CRCs, Lynch-like CRCs do not show epigenetic inactivation of the *MLH1* gene or mutations in BRAF. About 50-60% of CRCs in Lynch-like syndrome patients show biallelic inactivation at the somatic level of the MMR DNA genes within the tumor [83,84].

Finally, it is also possible that patients with Lynch-like syndrome have germline mutations in genes other than DNA MMR genes, which are known to be associated with the molecular mechanism that leads to LS. For example, germline mutations in *POLD1*/*POLE* DNA polymerases are found in Lynch-like syndrome patients [85].

## 11. Conclusions

In this review, we have outlined the main points of LS. Unfortunately, a large amount of variants identified in MMR genes remain of uncertain significance. This places a major limit on the molecular diagnosis of LS. Therefore, many individuals with a clinical suspicion of LS receive a diagnosis of Lynch-like syndrome. It is hoped that, in the near future, advances in high-throughput technologies and bioinformatic studies will improve the interpretation of these variants and provide important insights into the development of the disease, which will allow us to better classify family risk and related approaches to clinical surveillance and pave the way toward increasingly personalized therapeutic approaches for LS.

## Figures and Tables

**Figure 1 cancers-15-00075-f001:**
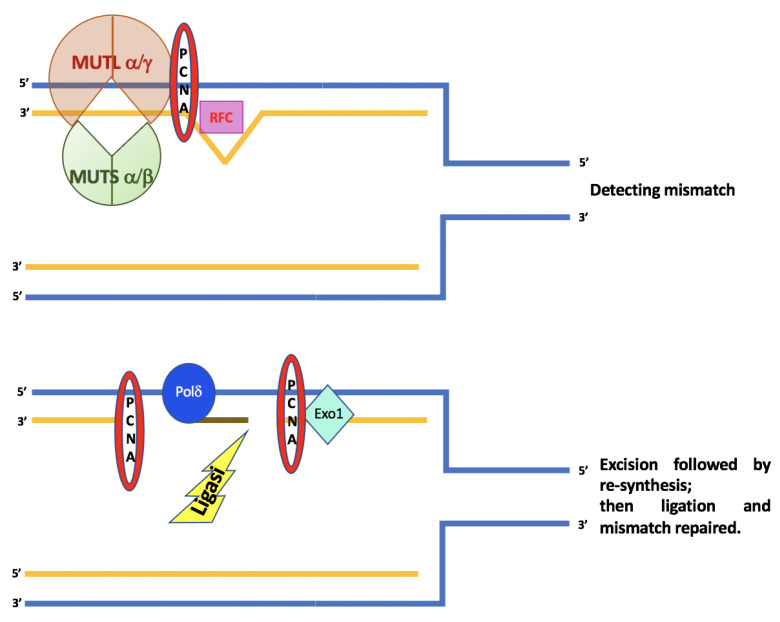
A schematic diagram for mechanisms and functions of human DNA mismatch repair.

**Figure 2 cancers-15-00075-f002:**
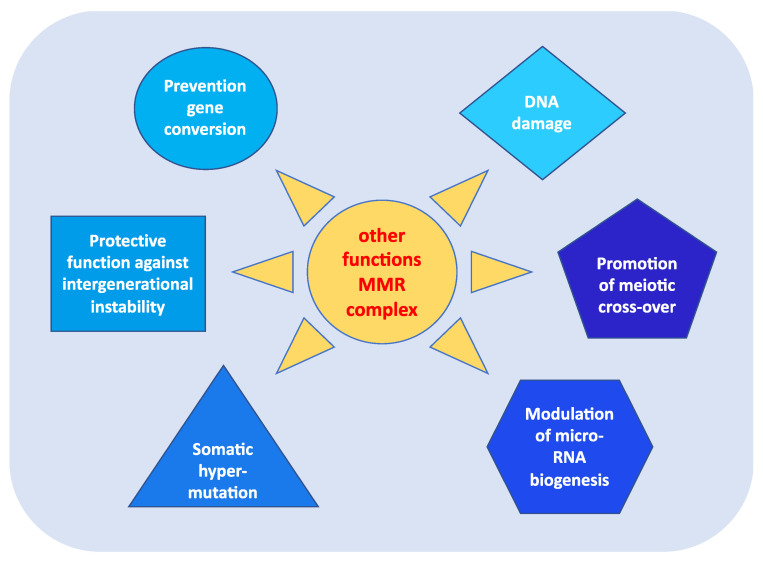
Other functions of MMR complex.

**Figure 3 cancers-15-00075-f003:**
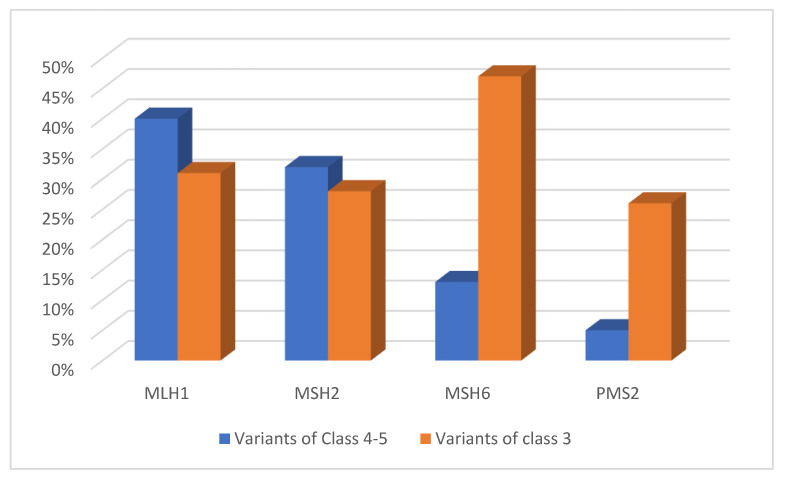
Distribution of pathogenic and likely pathogenic variants and VUS in MMR repair genes.

**Figure 4 cancers-15-00075-f004:**
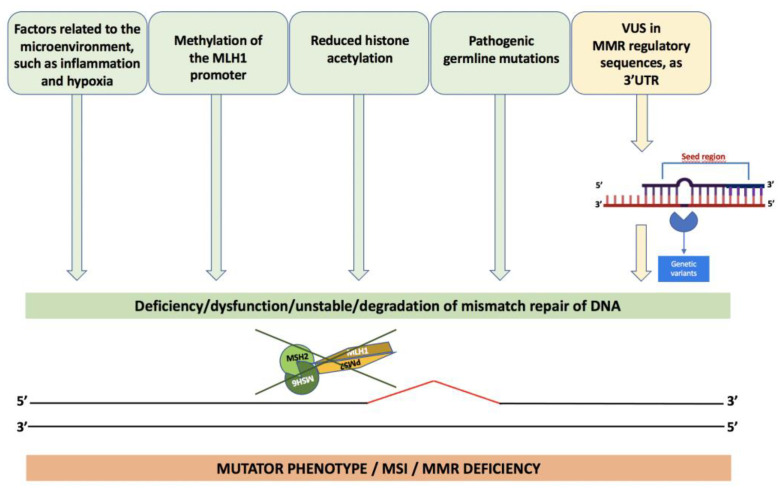
Causes of the loss of expression of MMR genes.

**Table 1 cancers-15-00075-t001:** Hereditary syndromes with known genetic predisposition to CRC.

Syndrome	Genes	Hereditary	Incidence	Lifetime crc Risk
Lynch Syndrome LS	MLH1, MSH2, MSH6, PMS2, EPCAM	AD	3–5%	15–90%
Familial Adenomatous Polyposis FAP	APC	AD	1%	Classic forms 100%; Attenuated forms until 70%
*Mutyh*-Associated Polyposis MAP	MUTYH	AR	1%	43–99%
Peutz-jeghers Syndrome PJS	STK11	AD	<1%	39%
Juvenile Polyposis JPS	SMAD4, BMPR1A	AD	<1%	39–69%

**Table 2 cancers-15-00075-t002:** MMR proteins and their functions.

*Bacterial MMR System*	*Yeast MMR System*	*Human MMR System*	Functions
MutS	MutS*α* (MSH2/MSH6) MutS*β* (MSH2/MSH3)	MutSα (MSH2/MSH6) MutS*β* (MSH2/MSH3)	Mismatch recognition
MUTL	MutL*α* (MLH1/PMS1) MutL*β* (MLH1/MLH2) MutLγ (MLH1/MLH3)	MutL*α* (MLH1/PMS2) MutL*β* (MLH1/PMS1) MutL*γ* (MLH1/MLH3)	Match making
MutH	PCNA	PCNA	Strand incision
	RFC	RFC	
	MutL*α* (MLH1/PMS1) MutL*γ* (MLH1/MLH3)	MutL*α* (MLH1/PMS2) MutL*γ* (MLH1/MLH3)	
RecJ	EXO1	EXO1	Strand excision (exonuclease)
ExoI			
ExoVII			
ExoX			
UvrD	-	-	Strand excision (helicase)
DNA polymerase III	DNA polymerase *δ*	DNA polymerase *δ*	Repair synthesis

**Table 3 cancers-15-00075-t003:** Strategies used to clarify the clinical role of VUS.

Direct Genetic Evidence	Indirect Genetic Evidence
Co-segregation with the disease	In silico predictions based on the position and nature of the amminoacid change
Co-occurrence with known pathogenic mutations	In vitro functional assays
Comparison of allele frequency in cases and controls	Biochemical assays for protein characterization
	LOH and methylation analysis in vivo

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
