# Peer review of "Hereditary Colorectal Cancer: State of the Art in Lynch Syndrome"

_cancers, 2022, doi:10.3390/cancers15010075_

Round 1
Reviewer 1 Report
1. An English-speaking editor is needed.
2. Lane 55: I think the following statement is not necessary “A systematic literature search was conducted in the literature databases PUBMED/Medline, Scopus and Google” as this is the way how reviews are prepared in general.
3. Lane 60 “the info." I suggest using more formal language: information instead of info.
4. Table 1, information regarding FAP in the last column entitled "Lifetime risk of CRC" is unclear: "attenuate forms until to 70%". What does it mean?
5. Lane 72 “MSH2 protein is therefore essential for the functional constitution of both complexes." However, the two sentences before do not lead to this conclusion. Instead, they point out the differences in the function of the two MMR complexes.
6. The standard rule is first to spell out the names and then provide appropriate abbreviations. Next, the authors need to scan the entire manuscript and fix it. An example is: "PCNA (proliferating cell nuclear antigen) factor."
7. The statement “Activation of the MutSα complex is characterized by an ATPase activity" should be reconsidered as activation characterized by activity does not make sense.
8. Lane 85: Which part of the complex undergoes a conformational change. "leading to a conformational change." A detailed description should be written in a more precise way and should be accompanied by figure to help the readers to understand the MMR process better.
9. In the section "2. MMR genes," the author randomly switch between human and bacterial names of the complexes. This needs to be clarified. I understand that authors are trying to point out homology between different species; however, something else is required here. Therefore, I suggest using only names of human proteins/complexes in the main text as LS is a human disease and providing a table with bacterial/yeast/human terms of proteins for readers.
10. Please refer to the nomenclature (e.g., https://www.jci.org/kiosk/publish/genestyle)
11. Lane 156 “minor genes." There are no minor genes.
12. Lane 166-167: “Most mutations that fall into MLH1, MSH2, and MSH6 are truncating, predominantly nonsense, or frameshift mutations. The proportion of missense mutations leading to single amino acid substitutions is between 30–60% for all four genes". The authors mentioned four genes in the second sentence, but only three are listed in the previous sentence.
13. The section "4. The molecular alteration of LS". First, describe the most common mutations and then move on to not frequently mutated genes. Currently, the authors talk about genes that are commonly mutated, then genes that are less frequently mutated, and back to genes that are frequently mutated. This generates unnecessary confusion.
14. A clear description of MSI-H and MSI-L with the frequency of mutations should be provided.
15. A table comparing LS and LS-like syndromes would be very helpful.
16. Some sentences are too long and as such do not provide clear information, e.g., "Therefore, recent studies have shown that a significant percentage of hereditary susceptibility related to the most common diseases may be the result of an additive effect of a series of low-frequency variants of different genes, which probably act in a dominant and independent way, giving each of these a moderate, but detectable, increase in the relative tumor risk [30-31,42-43].”
17. The authors use “loss of MMR proteins” and “Loss of expression of MMR genes". However, it needs to be clarified from the text that authors use them to mark different levels of genetic changes.
18. Figure 3. The standard rule is to present DNA with a 5' to 3' direction as the top strand, not the bottom. This figure needs to provide more helpful information. If authors would like to demonstrate different levels of regulation and, thus, possible levels of dysregulation of the MMR system, they should depict it.
19. There is a variation in the depiction of the references in the text () versus []. Please use the appropriate endnote style.
Author Response
REVIEWER 1
An English-speaking editor is needed.
Response 1. We thank you for the suggestion and attach the English language revision certificate to this new submission.
- Lane 55: I think the following statement is not necessary “A systematic literature search was conducted in the literature databases PUBMED/Medline, Scopus and Google” as this is the way how reviews are prepared in general.
Response 2. We took the reviewer comment and we deleted the sentence from the manuscript, at lane 70.
- Lane 60 “the info." I suggest using more formal language: information instead of info.
Response 3. We took the reviewer comment and we corrected in the text, at lane 74.
- Table 1, information regarding FAP in the last column entitled "Lifetime risk of CRC" is unclear: "attenuate forms until to 70%". What does it mean?
Response 4. We apologize for this typing error, we fixed this error in the revised version of the manuscript, “attenuated forms”.
- Lane 72 “MSH2 protein is therefore essential for the functional constitution of both complexes." However, the two sentences before do not lead to this conclusion. Instead, they point out the differences in the function of the two MMR complexes.
Response 5. We took the reviewer comment and we corrected in the text, as” The first complex is able to recognize …... Instead, the second complex is responsible …... MSH2 protein is essential for the functional constitution of both complexes”, at lanes 97-100.
- The standard rule is first to spell out the names and then provide appropriate abbreviations. Next, the authors need to scan the entire manuscript and fix it. An example is: "PCNA (proliferating cell nuclear antigen) factor."
Response 6. We apologize for these typing error, we fixed these error in the revised version of the manuscript, at lane 109.
- The statement “Activation of the MutSα complex is characterized by an ATPase activity" should be reconsidered as activation characterized by activity does not make sense.
Response 7. We took the reviewer comment and we corrected in the text, as ….“ The ATPase activity of the MutSα complex is important for the interaction with the unpaired DNA and the initiation of repair activity”, at lanes 111-112.
- Lane 85: Which part of the complex undergoes a conformational change. "leading to a conformational change." A detailed description should be written in a more precise way and should be accompanied by figure to help the readers to understand the MMR process better.
Response 8. We took the reviewer comment and we provided the figure in the text, the figure 1.
- In the section "2. MMR genes," the author randomly switch between human and bacterial names of the complexes. This needs to be clarified. I understand that authors are trying to point out homology between different species; however, something else is required here. Therefore, I suggest using only names of human proteins/complexes in the main text as LS is a human disease and providing a table with bacterial/yeast/human terms of proteins for readers.
Response 9. We took the reviewer comment, we corrected the text, as required and we provided to insert a table, the table 2.
- Please refer to the nomenclature (e.g., https://www.jci.org/kiosk/publish/genestyle)
Response 10. We apologize for these typing error, we fixed these error in the revised version of the manuscript.
- Lane 156 “minor genes." There are no minor genes.
Response 11. We took the reviewer comment and we corrected in the text, as: “with regard to other MMR gene”, at lane 186.
- Lane 166-167: “Most mutations that fall into MLH1, MSH2, and MSH6 are truncating, predominantly nonsense, or frameshift mutations. The proportion of missense mutations leading to single amino acid substitutions is between 30–60% for all four genes". The authors mentioned four genes in the second sentence, but only three are listed in the previous sentence.
Response 12. We apologize for these typing error, we fixed these error in the revised version of the manuscript, we added also the PMS2 gene, at lane 192.
- The section "4. The molecular alteration of LS". First, describe the most common mutations and then move on to not frequently mutated genes. Currently, the authors talk about genes that are commonly mutated, then genes that are less frequently mutated, and back to genes that are frequently mutated. This generates unnecessary confusion.
Response 13. We took the reviewer comment and we tried to correct the text as required, at lanes 191-200.
- A clear description of MSI-H and MSI-L with the frequency of mutations should be provided.
Response 14. We took the reviewer comment and we added the sentence in the text, lanes 270-274.
- A table comparing LS and LS-like syndromes would be very helpful.
Response 15. We thank the reviewer for the suggestion, however in the text all the differences that are highlighted between the two forms are described, therefore the table would only repeat the information reported in the text.
- Some sentences are too long and as such do not provide clear information, e.g., "Therefore, recent studies have shown that a significant percentage of hereditary susceptibility related to the most common diseases may be the result of an additive effect of a series of low-frequency variants of different genes, which probably act in a dominant and independent way, giving each of these a moderate, but detectable, increase in the relative tumor risk [30-31,42-43].”
Response 16. We took the reviewer comment and we tried to correct the text as required, at lanes 234-238.
- The authors use “loss of MMR proteins” and “Loss of expression of MMR genes". However, it needs to be clarified from the text that authors use them to mark different levels of genetic changes.
Response 17. We took the reviewer comment and we added the sentence in the text, as:….at somatic level, at lane 335.
- Figure 3. The standard rule is to present DNA with a 5' to 3' direction as the top strand, not the bottom. This figure needs to provide more helpful information. If authors would like to demonstrate different levels of regulation and, thus, possible levels of dysregulation of the MMR system, they should depict it.
Response 18. We apologize for these typing error, we fixed these error in the revised version of the manuscript, fig.4.
- There is a variation in the depiction of the references in the text () versus []. Please use the appropriate endnote style.
Response 19. We apologize for these typing errors, we fixed these error in the revised version of the manuscript.
Thank you very much for your suggestion, we provided to perform the revisions requested.

Reviewer 2 Report
This is overall an interesting paper. This can be an informative paper if the authors can improve this.
The authors need to check the text and references carefully. I point out just a few here, but that’s not all.
The second sentence “LS is 27 also known as Hereditary Non-Polyposis Colorectal Cancer (HNPCC) to highlight the 28 absence of colon polyps and distinguish it from FAP, which is characterized by 100-1,000 29 polyps [3], as well as other inherited syndromes such as hamartomatous polyposis [4] 30 Tab.1.”
Is clearly incorrect. They should write “… absence of colon polyposis and…”. Clearly HNPCC and LS patients can have polyps.
Ref. 1 is incorrectly cited. That’s not relevant to the incidence of LS.
3-5% of LS among CRC patients is an overestimate, likely based on data from tertiary-care hospital series. A recent meta-analysis by Abu-Ghazaleh et al. in Genet Med 2022 showed 2-3% estimate of LS among all CRC patients. I believe that is a more correct estimate than 3-5%.
Refs. 2-3 may not be appropriate for the incidence of FAP in the world.
MUTYH biallelic inactivation can cause a distinct pattern of mutations different from classical mismatch repair deficiency (Georgeson et al. Nat Commun 2022). The authors should discuss that.
LS causes early-onset CRC but LS is not the reason of the recent rise of early-onset CRC, which is discussed in detail by Akimoto et al. (Nat Rev Clin Oncol 2021). This fact should be discussed.
Author Response
REVIEWER 2
This is overall an interesting paper. This can be an informative paper if the authors can improve this.
The authors need to check the text and references carefully. I point out just a few here, but that’s not all.
The second sentence “LS is 27 also known as Hereditary Non-Polyposis Colorectal Cancer (HNPCC) to highlight the 28 absence of colon polyps and distinguish it from FAP, which is characterized by 100-1,000 29 polyps [3], as well as other inherited syndromes such as hamartomatous polyposis [4] 30 Tab.1.”
Is clearly incorrect. They should write “… absence of colon polyposis and…”. Clearly HNPCC and LS patients can have polyps.
Response 1. We took the reviewer comment and we added the sentence in the text, lanes 42-44.
Ref. 1 is incorrectly cited. That’s not relevant to the incidence of LS. 3-5% of LS among CRC patients is an overestimate, likely based on data from tertiary-care hospital series. A recent meta-analysis by Abu-Ghazaleh et al. in Genet Med 2022 showed 2-3% estimate of LS among all CRC patients. I believe that is a more correct estimate than 3-5%.
Response 2. We took the reviewer comment, at lane 39, and we provided to update the reference 1.
Refs. 2-3 may not be appropriate for the incidence of FAP in the world.
Response 3. We apologize for this error and we provided to update the reference 2, instead we don’t change the reference 3 because is appropriate.
MUTYH biallelic inactivation can cause a distinct pattern of mutations different from classical mismatch repair deficiency (Georgeson et al. Nat Commun 2022). The authors should discuss that.
Response 4. We thank the reviewer for the suggestion, however no MAP related issues are addressed in this review so we have not filled in what was requested.
LS causes early-onset CRC but LS is not the reason of the recent rise of early-onset CRC, which is discussed in detail by Akimoto et al. (Nat Rev Clin Oncol 2021). This fact should be discussed.
Response 5. We took the reviewer comment and we added the sentence in the text, lanes 46-49.
Thank you very much for your suggestion, we provided to perform the revisions requested.

Reviewer 3 Report
The authors performed a literature search and summarized genetic and clinical features of Lynch syndrome. The manuscript is well written about genetic alteration and MMR proteins. The comparison in hereditary syndrome associated with CRC is shown in Table 1, but the table is ordinal. It can be deleted. Instead, a reviewer expect to read the progression in the genetic testing and CRC treatment in Lynch syndrome.
Author Response
REVIEWER 3
The authors performed a literature search and summarized genetic and clinical features of Lynch syndrome. The manuscript is well written about genetic alteration and MMR proteins.
The comparison in hereditary syndrome associated with CRC is shown in Table 1, but the table is ordinal. It can be deleted.
We thank the reviewer for the suggestion, however we have included table 1 to illustrate the other hereditary syndromes that predispose to colorectal cancer in addition to Lynch syndrome and that we have not reported in the text.
Instead, a reviewer expect to read the progression in the genetic testing and CRC treatment in Lynch syndrome.
We thank the reviewer for the suggestion, however we have not included in the manuscript the progress regarding the treatment of CRC in Lynch syndrome because we have tried to give particular attention to genetic testing to effect a significant prevention of the disease and in order to underlie the importance to study all those variants that remain of uncertain significance.

Reviewer 4 Report
The manuscript titled "Hereditary colorectal cancer: State of the art in Lynch syndrome" (cancers-2083250) submitted to Cancers, describes a very interesting and useful approach describing Lynch syndrome. Authors presented recent advances on the topic and in particular described topic close most recent information about the several causes that can lead to the loss expression of MMR genes. Review is very well organized and written. Manuscript is divided properly and after short introduction, authors in very comprehensive way described MMR genes with list of functions, next the molecular alterations of Lynch syndrome and . variants of uncertain significance and microsatellite instability (MSI) and so on. Review is based on almost 85 references, concluded that too many variants identified in MMR genes remain of uncertain significance and hopefully advances in high-throughput technologies and bioinformatics studies will improve the interpretation of these variants and also provide important insights into the development of the disease. To sum up in my opinion manuscript is written in correct way, and it could be published in present form without any changes.
Author Response
REVIEWER 4
The manuscript titled "Hereditary colorectal cancer: State of the art in Lynch syndrome" (cancers-2083250) submitted to Cancers, describes a very interesting and useful approach describing Lynch syndrome. Authors presented recent advances on the topic and in particular described topic close most recent information about the several causes that can lead to the loss expression of MMR genes. Review is very well organized and written. Manuscript is divided properly and after short introduction, authors in very comprehensive way described MMR genes with list of functions, next the molecular alterations of Lynch syndrome and . variants of uncertain significance and microsatellite instability (MSI) and so on. Review is based on almost 85 references, concluded that too many variants identified in MMR genes remain of uncertain significance and hopefully advances in high-throughput technologies and bioinformatics studies will improve the interpretation of these variants and also provide important insights into the development of the disease. To sum up in my opinion manuscript is written in correct way, and it could be published in present form without any changes.
We thank the reviewer for the positive comments on our manuscript.

Round 2
Reviewer 1 Report
The authors have satisfactorily addressed most of my concerns.